# Disparities in COVID-19 Vaccination among Low-, Middle-, and High-Income Countries: The Mediating Role of Vaccination Policy

**DOI:** 10.3390/vaccines9080905

**Published:** 2021-08-14

**Authors:** Yuqi Duan, Junyi Shi, Zongbin Wang, Shuduo Zhou, Yinzi Jin, Zhi-Jie Zheng

**Affiliations:** 1Department of Global Health, School of Public Health, Peking University, Beijing 100191, China; 18811318259@163.com (Y.D.); shijunyi_2009@163.com (J.S.); wangzongbin@bjmu.edu.cn (Z.W.); zhoushuduo@pku.edu.cn (S.Z.); zhengzj@bjmu.edu.cn (Z.-J.Z.); 2Institute for Global Health and Development, Peking University, Beijing 100191, China

**Keywords:** COVID-19 vaccine, equitable access, income level, policy strength, mediation effect

## Abstract

Inequity in the access to and deployment of the coronavirus disease 2019 (COVID-19) vaccines has brought about great challenges in terms of resolving the pandemic. Aiming to analyze the association between country income level and COVID-19 vaccination coverage and explore the mediating role of vaccination policy, we conducted a cross-sectional ecological study. The dependent variable was COVID-19 vaccination coverage in 138 countries as of May 31, 2021. A single-mediator model based on structural equation modeling was developed to analyze mediation effects in different country income groups. Compared with high-income countries, upper-middle- (*β* = −1.44, 95% CI: −1.86–−1.02, *p* < 0.001), lower-middle- (*β* = −2.24, 95% CI: −2.67–−1.82, *p* < 0.001), and low- (*β* = −4.05, 95% CI: −4.59–−3.51, *p* < 0.001) income countries had lower vaccination coverage. Vaccination policies mediated 14.6% and 15.6% of the effect in upper-middle- (*β* = −0.21, 95% CI: −0.39–−0.03, *p* = 0.020) and lower-middle- (*β* = −0.35, 95% CI: −0.56–−0.13, *p* = 0.002) income countries, respectively, whereas the mediation effect was not significant in low-income countries (*β* = −0.21, 95% CI: −0.43–0.01, *p* = 0.062). The results were similar after adjusting for demographic structure and underlying health conditions. Income disparity remains an important cause of vaccine inequity, and the tendency toward “vaccine nationalism” restricts the functioning of the global vaccine allocation framework. Stronger mechanisms are needed to foster countries’ political will to promote vaccine equity.

## 1. Introduction

The coronavirus disease 2019 (COVID-19) pandemic has infected more than 170 million people, caused more than 3.7 million deaths [1], and had a severe negative impact on the global economy. Rapid development, allocation, and deployment of safe and effective vaccines is the key instrument to save lives and contain the pandemic. To this end, there has been an unprecedented level of global funding and collaboration for COVID-19 vaccine development. As of early June 2021, there were 322 vaccine candidates worldwide, of which 97 were in clinical testing and 17 were in use [2]. However, barriers to the equitable allocation and deployment of the vaccines present a significant challenge to accelerating the end of the pandemic. At present, approximately 20% of the global population has received at least one dose of a COVID-19 vaccine, but this number is less than 1% in low-income countries [3].

The equitable allocation of vaccines globally means that all countries, regardless of their developmental or economic status, should have equitable access to vaccines, as assessed by the objective possibility for countries to obtain vaccines. The disadvantageous position of low- and middle-income countries in accessing scarce resources such as vaccines was widely noted during the highly pathogenic avian influenza A (H5N1) epidemic in 2004 and the influenza A (H1N1) pandemic in 2009 [4,5]. To address the issue of vaccine access in low- and middle-income countries, observed in previous pandemics, during the COVID-19 pandemic, the World Health Organization led the launch of COVID-19 Vaccines Global Access (COVAX) as an implementation framework for the equitable allocation of and access to COVID-19 vaccines [6].

At present, at the country level, nations are accessing vaccines through several approaches, mainly including advance purchase agreements with manufacturers, bilateral agreements between countries, regional procurement arrangements, and COVAX. However, COVAX seems to be insufficient to reverse the inequitable access to the vaccines, and a country’s income level is still a key factor in determining vaccine access. As early as the first half of 2020, some high- and middle-income countries had already purchased enough vaccine doses through advance purchase agreements. The first purchases for low-income countries came in January 2021 through the African Union’s pooled procurement approach. Some high-income countries have procured enough vaccines to cover their populations five times over, whereas low- and middle-income countries generally do not have enough doses [7].

At the country level, vaccine deployment refers to the development and implementation of country vaccination policies based on the global allocation process and country access to vaccines. Several studies extensively explored the prioritization of vaccination on the basis of the principles of equity, ethics, and effectiveness [8,9,10,11,12]. The framework for the equitable allocation of COVID-19 vaccines recommended by World Health Organization also offers an overarching structure for vaccine deployment to assist policy makers in individual countries. This framework suggests a two-phase approach. In the first phase, countries should prioritize the vaccination of key populations such as health and social care workers, older adults, and adults with comorbidities, together accounting for approximately 20% of their population. In the second phase, after all countries have completed the first phase, additional populations should be covered, considering factors such as COVID-19 threat and country vulnerability [13]. However, there are large gaps in the process of implementing vaccination policies across individual countries.

In summary, under current global allocation framework of vaccines, a country’s income levels will affect its vaccine access, which in turn affects its vaccine deployment and ultimately vaccination coverage. Previous studies on COVID-19 vaccine equity have mostly focused on vaccine access in countries with different income levels, without considering country-level vaccine deployment (i.e., domestic vaccination policies). This study analyzed the impact of national income level on vaccination policies and vaccination coverage and explored the mediating role of vaccination policies in the relationship between a country’s income level and vaccination coverage. Therefore, we can break down the different stages of national vaccine access and deployment to deeply explore the factors that influence vaccination coverage.

## 2. Materials and Methods

### 2.1. Study Design and Data

We conducted a cross-sectional study linking national income level with COVID-19 vaccination policy and vaccination coverage across multiple countries. All data for this study were obtained from open and publicly available data sources. Specifically, the data on vaccination coverage were obtained from Our World in Data [3], which compiles the most recent vaccination data from governments and health departments around the world. Data on vaccination policies were obtained from the Oxford COVID-19 Government Response Tracker [14], which provides a simple way to calculate the strength of country vaccination policies using information pooled from various sources. For these two datasets, the latest information as of 31 May 2021, was selected. Considering the large variation in the timing of vaccination coverage data updates in different country contexts, countries for which vaccination coverage data were not available from 25 May to 31 May 2021, were excluded to ensure comparability. Data on country income levels and demographic structure were obtained from the World Bank [15], and data on underlying health conditions were obtained from the Institute for Health Metrics and Evaluation [16]; both datasets provided data from 2019. We use these data for its reliability and consistent methods, which were widely used in scientific research [17,18,19,20,21]. We matched these variables, built a database, and excluded countries with missing values on any variable of interest. Ultimately, 138 countries were included in the analysis.

### 2.2. Measures

The dependent variable was country-level COVID-19 vaccination coverage. Because of differences in vaccine doses and data collection limitations, the actual vaccination coverage—the proportion of people partially or fully vaccinated—was not available for some countries. Therefore, the total number of vaccination doses administered per 100 people was used to represent the vaccination coverage in each country.

The independent variable was the income level of each country. This was mainly measured by the gross national income per capita in current United States dollars (USD) (using the World Bank’s Atlas method exchange rates) in 2019 for each country, and all included countries were categorized as high-, upper-middle-, lower-middle-, or low-income, with reference to the World Bank’s income group categories for 2021 [22]. Among 138 countries included for analysis, there were 51 high-income countries, 36 upper-middle-income countries, 34 lower-middle-income countries, and 17 low-income countries.

The mediating variable was the strength of each country’s vaccination policy, which incorporated the population covered by the country vaccination policy, as well as vaccine affordability. For better interpretation, we further converted this variable to a score of 1–6 on the basis of the original percentage scale, where a score of 1 represented the lowest policy strength, reflecting low availability, and a score of 6 represented the highest policy strength (Appendix A Table A1) [23].

In addition, to adjust for the effects of demographic structure and underlying health conditions on vaccination coverage, the percentage of the population ages 65 and above, the prevalence of cardiovascular diseases (per 100 people), the prevalence of chronic respiratory diseases (per 100 people), and the prevalence of diabetes mellitus (per 100 people) for each country were included as covariates (Appendix A Table A2).

### 2.3. Statistical Analysis

Descriptive statistics were used to determine differences in average vaccination coverage, vaccination policy strength, demographic structure, and underlying health conditions across countries with different income levels. We developed a single-mediator model to determine the effect of country income level (X) on COVID-19 vaccination coverage (Y), as well as the mediating role of vaccination policy strength (M) in the impact pathway. To test the robustness of the model, we estimated Model 1 without covariates and Model 2 with partial constraints, controlling for demographic structure and underlying health conditions (C) (Figure 1).

To quantify the association, we used a series of linear regression models. X was a categorical variable (with the high-income countries as the reference group). Y was a continuous variable after logarithmic transformation. Thus, we analyzed the relative mediation effects of different country-level income groups. For each group, we estimated the relative indirect effect, which was given by the product of the coefficient of the association between (a) the independent variable and the mediator and (b) the mediator and the dependent variable. The correlation of path a represents the change in vaccination policy strength in countries with other income levels compared with high-income countries; the correlation coefficient of path b indicates the degree of change in national vaccination coverage for each one-point of increase in policy strength. The product of a and b, and the total effect coefficient, respectively, represent how country income level influences vaccination coverage by influencing vaccination policy strength and in general. We used the ratio of the relative indirect effect to the relative total effect to calculate the percentage mediated—the percentage of the independent variable–dependent variable association that can be explained by the mediator [24,25].

Analysis of variance was used for the descriptive statistics, and structural equation modeling was used to analyze the mediation effects. All analyses were performed using Stata 14.0 (StataCorp, College Station, TX, USA), and 2-sided *p* < 0.05 was considered statistically significant.

## 3. Results

Among the 138 included countries, higher proportions of the population aged ≥ 65 years and with underlying diseases including cardiovascular diseases, chronic respiratory diseases, and diabetes mellitus were found in high-income countries, as were higher vaccination policy scores (*p* < 0.001). Vaccination coverage varied widely by income group, ranging from 0.03 doses/100 people to 136.74 doses/100 people, with average coverage of 58.49 doses/100 people, 17.30 doses/100 people, 11.95 doses/100 people, and 1.26 doses/100 people for high-, upper-middle-, lower-middle-, and low-income countries, respectively (Table 1).

Although most countries have implemented vaccination policies of moderate strength, in Figure 2, the green dots representing high policy strength tended to be concentrated in the upper right corner of the plot, suggesting that countries with higher income levels have more extensive vaccine policies, as well as higher vaccination coverage, whereas the red dots, representing low policy strength, were concentrated in the lower middle area of the plot, suggesting that countries with lower income levels have more limited vaccination policies and lower vaccination coverage (Figure 2).

In the regression analysis, the mediation models showed that income level was significantly associated with vaccination coverage. Vaccination policy strength was significantly associated with income level and vaccination coverage, suggesting that vaccination policy strength was a mediator of country-level socioeconomic vaccination disparities. Specifically, compared with high-income countries, vaccination policy strength was lower in upper-middle- (*β* = −0.64, *p* = 0.007), lower-middle- (*β* = −1.04, *p* < 0.001), and low- (*β* = −0.63, *p* = 0.042) income countries. Lower policy strength was associated with lower vaccination coverage (*β* = 0.33, *p* < 0.001). Compared with high-income countries, upper-middle- (*β* = −1.44, *p* < 0.001), lower-middle- (*β* = −2.24, *p* < 0.001), and low- (*β* = −4.05, *p* < 0.001) income countries had lower vaccination coverage. Vaccination policy strength explained 14.6% and 15.6% of the association between income level and vaccination coverage in upper-middle- (*β* = −0.21, *p* = 0.020) and lower-middle- (*β* = −0.35, *p* = 0.002) income countries, respectively. However, the mediation effect was not significant (*β* = −0.21, *p* = 0.062) in low-income countries. Similar results were observed after adjusting for demographic structure and underlying health conditions, demonstrating the robustness of the model (Table 2).

## 4. Discussion

Ensuring equitable access to and deployment of vaccines is crucial for containing the COVID-19 pandemic. Six months after the development, allocation, and deployment of COVID-19 vaccines, globally, vaccination coverage remains lower in countries with lower income levels, demonstrating that additional efforts are needed to narrow the gap in vaccination coverage across countries.

Consistent with the results of previous studies, this study has shown that national income level is an important factor influencing country-level vaccine access, vaccination policy, and vaccination coverage. Prior to widespread vaccine allocation by COVAX, a large number of studies expressed concern about the fact that the majority of global vaccine purchases and vaccinations were occurring in a few high-income countries [7,26,27]. A study analyzing the relationship between macro-socioeconomic factors and the global allocation of COVID-19 vaccines showed that higher gross domestic product per capita was associated with larger numbers of vaccinations [28], and similar findings were also observed within the United States [29]. A few studies focused on the different stages of vaccination policies in different countries, which showed that the poorest countries were often lagging behind in with clear strategies or resources to promote COVID-19 vaccination [30]. However, there is still a lack of research to develop the relationship between national economic level, vaccination policy and vaccination coverage.

The present findings suggest that in general, countries with higher income levels have taken the lead in implementing more comprehensive vaccination policies and have higher vaccination coverage. These outcomes depend on a country’s ability to access vaccines and their willingness to deploy them. High-income countries have invested heavily in vaccine development and procurement and thus have an advantage in prioritization for vaccine access. In contrast, most low- and middle-income countries need to compete for vaccines on the open market, although some countries can access vaccine commitments by establishing production and clinical trial collaborations with manufactures [12,31]. Countries with vaccine reserves tend to prioritize their own vaccination processes [32] and thus have higher vaccination coverage, to some extent reflecting the phenomenon of “vaccine nationalism.”

The above phenomena reflect the limitations of the current global vaccine allocation mechanism. First, COVAX cannot currently integrate global vaccine procurement channels. Countries with the ability to access vaccines are still procuring them through advance purchase agreements with manufactures, making COVAX only one of many competitors in the global vaccine market and rendering it unable to secure sufficient vaccine doses. Second, for the doses already obtained by COVAX, practical problems associated with transporting and deploying the vaccines have forced the World Health Organization to prioritize countries with adequate vaccine deployment infrastructure, which undermines the principle that countries should receive vaccine doses at the same rate [33]. In addition, gaps in financial commitments to COVAX and uncertainty regarding vaccine supplies also threaten the effective implementation of the global vaccine allocation mechanism [12].

This study has shown that the process of vaccine deployment within individual countries may be a direct way in which income level affects vaccination coverage. Countries with lower income levels often lack stable vaccination management and information systems, functional cold chains, transport and public health infrastructure, adequate service providers, and financial support [34], all of which can affect the rapid deployment of vaccines. However, several studies have also shown that vaccine acceptance is lower among low-income populations because of a lack of accurate knowledge about COVID-19 risks and vaccine effectiveness [35,36,37,38,39]. In particular, vaccination reluctance among 65+ age group in the emerging and low-income group is more noteworthy. Vaccine deployment at the country level and acceptance at the individual level may be influencing factors that were not included in the model in this study, and differences in these factors may be part of the reason vaccine policies did not show a significant mediation effect in low-income countries.

Based on previous studies [35,36,37,38,39], vaccine acceptance tends to be lower in low-income countries because of a lack of knowledge about disease risk and the effectiveness of vaccines. It is warranted to expand on their ideas of how to heighten knowledge about COVID-19 risk as well as vaccine effectiveness. Countries need to make efforts to shape scientific awareness and attitudes toward COVID-19 vaccines. Behavioral demonstration by cultural or public health leaders, health education by primary care personnel would make a difference. Furthermore, long-term efforts are still needed. Especially in the middle and late stages of global allocation of COVID-19 vaccines, the decreasing urgency of some high-income countries for their own vaccination needs may increase their willingness to promote vaccine equity. Strong advocacy by key actors in multilateral, cross-regional and regional mechanisms, as well as relatively binding benefit sharing and regulatory mechanisms will play a catalytic role. Strengthening vaccine deployment capacity in low- and middle-income countries should also be emphasized. On the basis of funding, equipment and technical support, it would be helpful to mobilize practice teams to help these countries in the last-mile deployment.

This study has several limitations. First, because of limited data availability, the data used in this study could only represent country-level vaccination coverage to a certain extent, and could not reflect the specific situation at the sub-national level. The lack of data from low-income countries may have affected the accuracy of the results. Second, because of the lack of specific data on vaccine deployment and vaccination acceptance, the model failed to incorporate multiple influencing factors, resulting in a simplification of the association. Finally, this was an ecological study and therefore could not assess causality. As global vaccination efforts continue, to provide direction for promoting vaccine equity, it is important to track the progress of equitable vaccine access and deployment and to further explore broader mediators, clarifying how income level affects vaccination coverage.

## 5. Conclusions

Globally, economic status remains a key factor influencing equitable vaccine access and deployment. High-income countries have priority access to more vaccine doses and tend to prioritize more extensive vaccination policies within their own countries, leading to higher vaccination coverage in these countries. There is a need to strengthen existing global vaccine allocation frameworks by considering the stringency of vaccine policy and capacity of vaccine deployment at the national level. Relatively binding benefit sharing and regulatory mechanisms are needed to promote nationwide vaccine deployment, and to bolster the political will of all countries to promote vaccine equity globally. Stronger mechanisms are also needed to help low-income countries address practical barriers to vaccine deployment.

## Figures and Tables

**Figure 1 vaccines-09-00905-f001:**
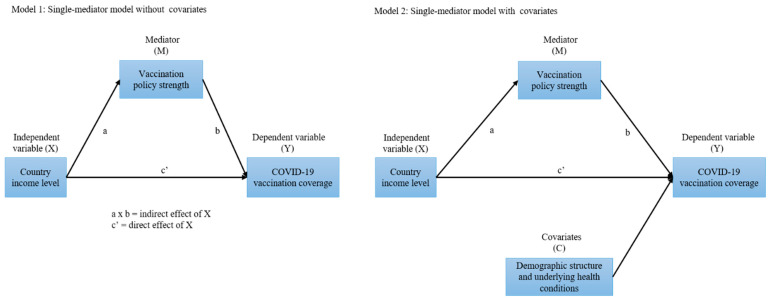
Schema for the mediation models. Note: COVID-19: coronavirus disease 2019. In Model 2, the covariates are the percentage of the population aged ≥65 years, the prevalence of cardiovascular diseases (per 100 people), the prevalence of chronic respiratory diseases (per 100 people), and the prevalence of diabetes mellitus (per 100 people).

**Figure 2 vaccines-09-00905-f002:**
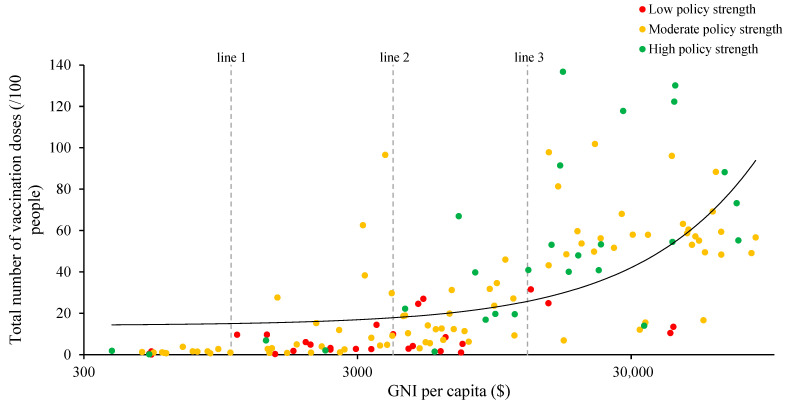
Comparison of COVID-19 vaccination coverage by country-level income and vaccination policy Strength. Note: Country-level income was classified on the basis of gross national income per capita in current USD (using the World Bank’s Atlas method exchange rates) in 2019 according to the World Bank. Line 1 separates lower-middle-income countries from low-income countries, line 2 separates upper-middle-income countries from lower-middle-income countries, and line 3 separates high-income countries from upper-middle-income countries. A vaccination policy strength score of ≤3 was defined as low policy strength, a score of >3 and ≤5 was defined as moderate policy strength, and a score of >5 was defined as high policy strength.

**Table 1 vaccines-09-00905-t001:** Characteristics of 138 countries around the world, stratified by income level.

Characteristics	Country-Level Income Groups, Mean (SD)				*p* Value
	High (*n* = 51)	Upper-Middle (*n* = 36)	Lower-Middle (*n* = 34)	Low (*n* = 17)	
GNI per capita, $	37226.08 (20079.89)	6934.72 (2123.69)	2502.94 (953.34)	657.65 (180.78)	<0.001
Age ≥ 65 years, %	15.95 (5.99)	8.84 (4.56)	5.37 (3.13)	2.88 (0.40)	<0.001
CVD prevalence, per 100 people	9.87 (3.12)	7.01 (3.02)	4.99 (2.03)	3.62 (0.42)	<0.001
CRD prevalence, per 100 people	8.99 (3.13)	5.65 (1.57)	4.38 (1.08)	4.36 (1.07)	<0.001
DM prevalence, per 100 people	9.11 (2.70)	6.83 (2.77)	4.30 (2.22)	2.28 (0.97)	<0.001
Vaccination policy strength	4.92 (0.93)	4.28 (1.21)	3.88 (1.23)	4.29 (1.21)	<0.001
Vaccination doses, per 100 people	58.49 (30.35)	17.30 (14.26)	11.95 (19.67)	1.26 (0.93)	<0.001

Note: Country-level income was classified on the basis of gross national income (GNI) per capita in current United States dollars (USD) (using the World Bank’s Atlas method exchange rates) in 2019 according to the World Bank. CVD: cardiovascular diseases; CRD: chronic respiratory diseases; DM: diabetes mellitus.

**Table 2 vaccines-09-00905-t002:** Mediation model: regression of country income level on COVID-19 vaccination coverage.

	Model 1: Without Covariates				Model 2: With Covariates			
	*β*	*p* Value	95% CI		*β*	*p* Value	95% CI	
			Lower	Upper			Lower	Upper
**Income level upper-middle vs. high**								
Path a: X→M	−0.64	0.007	−1.11	−0.17	−0.64	0.007	−1.11	−0.17
Path b: M→Y	0.33	<0.001	0.20	0.47	0.36	<0.001	0.23	0.50
Indirect Effect (a×b: X→M→Y)	−0.21	0.020	−0.39	−0.03	−0.23	0.017	−0.42	−0.04
Direct Effect (Path c’: X→Y adj M)	−1.22	<0.001	−1.62	−0.82	−0.74	0.003	−1.22	−0.26
Total Effect (Path c: X→Y)	−1.44	<0.001	−1.86	−1.02	−0.97	<0.001	−1.47	−0.47
% Mediated (a×b/c)	14.6%				23.7%			
**Income level lower-middle vs. high**								
Path a: X→M	−1.04	<0.001	−1.52	−0.56	−1.04	<0.001	−1.52	−0.56
Path b: M→Y	0.33	<0.001	0.20	0.47	0.36	<0.001	0.23	0.50
Indirect Effect (a×b: X→M→Y)	−0.35	0.002	−0.56	−0.13	−0.38	0.001	−0.60	−0.15
Direct Effect (Path c’: X→Y adj M)	−1.90	<0.001	−2.32	−1.48	−1.13	<0.001	−1.74	−0.52
Total Effect (Path c: X→Y)	−2.24	<0.001	−2.67	−1.82	−1.51	<0.001	−2.12	−0.90
% Mediated (a×b/c)	15.6%				25.2%			
**Income level low vs. high**								
Path a: X→M	−0.63	0.042	−1.23	−0.02	−0.63	0.042	−1.23	−0.02
Path b: M→Y	0.33	<0.001	0.20	0.47	0.36	<0.001	0.23	0.50
Indirect Effect (a×b: X→M→Y)	−0.21	0.062	−0.43	0.01	−0.23	0.058	−0.46	0.01
Direct Effect (Path c’: X→Y adj M)	−3.84	<0.001	−4.35	−3.33	−2.90	<0.001	−3.65	−2.15
Total Effect (Path c: X→Y)	−4.05	<0.001	−4.59	−3.51	−3.12	<0.001	−3.89	−2.35
% Mediated (a×b/c)	/				/			

Note: M = mediator (vaccination policy strength); X = independent variable (country income level); Y = dependent variable (vaccination doses per 100 people, log transformed); CI: confidence interval. The covariates in Model 2 include percentage of the population aged ≥ 65 years, the prevalence of cardiovascular diseases (per 100 people), the prevalence of chronic respiratory diseases (per 100 people), and the prevalence of diabetes mellitus (per 100 people).

## Data Availability

The datasets generated and analyzed during the current study are available in Our World in Data at https://ourworldindata.org/covid-vaccinations (accessed on 10 June 2021), the Oxford COVID-19 Government Response Tracker at https://www.bsg.ox.ac.uk/research/research-projects/COVID-19-government-response-tracker (accessed on 10 June 2021), the World Bank at https://data.worldbank.org/ (accessed on 10 June 2021), the Institute for Health Metrics and Evaluation at http://ghdx.healthdata.org/gbd-results-tool (accessed on 10 June 2021).

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
