# Peer review of "Disparities in COVID-19 Vaccination among Low-, Middle-, and High-Income Countries: The Mediating Role of Vaccination Policy"

_vaccines, 2021, doi:10.3390/vaccines9080905_

Round 1

Reviewer 1 Report

The authors provide an analysis of effects of income and vaccine mediation policy on the vaccination coverage. The authors employ a statistical regression approach. The novelty of the manuscript is extremely limited.

The authors use data from different sources just providing the reference of the source. They may be more detailed about the type of the data and the quality of the data.

There is hardly a comparison with other studies that addressed the same question. How do the estimates of the authors compare with the estimates of the other publications?

The type of the study and the statistical approach should be mentioned in the abstract.

Some more detailed comments.

Abstract:

For all estimates in the abstract 95% CI should be provided (for the regression coefficient as well as the effect of the vaccination policies.

Material and Methods

Please provide more information about the quality of the data.

Results:

The differences observed in values of the regression coefficients should be explained in relation to each other. The absolute values of the beta estimates are not directly interpretable for the reader. The authors should provide an interpretation of the beta values and a comparison.

Discussion

The quality of the data used and the implication of their crudeness should be discussed.

Author Response

 Dear editor,
We appreciate the opportunity to respond the comments from the reviewers for our manuscript. We have revised the manuscript accordingly. The following are the pointto-point responses to these comments of Reviewer 1.
Comments and Suggestions for Authors:

The authors provide an analysis of effects of income and vaccine mediation
policy on the vaccination coverage. The authors employ a statistical regression
approach. The novelty of the manuscript is extremely limited.

1. The authors use data from different sources just providing the reference of the
source. They may be more detailed about the type of the data and the quality of
the data.

Authors’ response:
Thanks for your valuable comments. In the revised manuscript
(tracks), we have added a table (Table S2) to better show the type of data in the
section of Appendix, line 337-342, page 9-10, as following:
Table S2. Variables evaluated for testing in mediation model

Variable Definition Type Source Date of Data
Update
Dependent variable
COVID-19 vaccination
coverage
The total number of
vaccination doses
administered per 100
people at the country
level
Continuous
variable
Our World
in Data
From May 25
to 31, 2021
Independent variable

Country income level Four country
categories including
high-, upper-middle-,
lower-middle-, or
low-income
Categorical
variable
The World
Bank
2021
Mediation variable
Vaccination policy
strength
A score of 1–6
(lowest-highest
strength), which
incorporated the
population covered by
the country
vaccination policy, as
well as vaccine
affordability at the
country level
Continuous
variable
The Oxford
COVID-19
Government
Response
Tracker
From May 25
to 31, 2021
Covariates
Age 65 years The percentage of the
population ages 65
and above at the
country level
Continuous
variable
The World
Bank
2019
CVD prevalence The prevalence of
cardiovascular
diseases per 100
people at the country
level
Continuous
variable
The
Institute for
Health
Metrics and
Evaluation
2019
CRD prevalence The prevalence of
chronic respiratory
diseases per 100
people at the country
level
Continuous
variable
The
Institute for
Health
Metrics and
Evaluation
2019
DM prevalence The prevalence of
diabetes mellitus per
100 people at the
country level
Continuous
variable
The
Institute for
Health
Metrics and
Evaluation
2019

Note: The dependent variable and mediation variable used the latest data from May 25 to 31, 2021. The independent variable was mainly measured by the gross national income per capita in current United States dollars (using the World Bank’s Atlas method exchange rates) in 2019, with reference to the World Bank’s income group categories for 2021. CVD: cardiovascular diseases; CRD: chronic respiratory diseases; DM: diabetes mellitus.
Moreover, in the revised manuscript (tracks), we have added the statement to reflect the quality of the data and references to evidence the data were widely used in scientific research in the section of Materials and Methods, line 106-107, page 3, as following:

“We use these data for its reliability and consistent methods, which were widelyused in scientific research [17-21].”

17. Mathieu, E.; Ritchie, H.; Ortiz-Ospina, E.; Roser, M.; Hasell, J.; Appel, C.;
Giattino, C.; Rodés-Guirao, L. A global database of COVID-19 vaccinations.
Nature Human Behaviour 2021, 5, 947-953, doi:10.1038/s41562-021-01122-8.
18. Hale, T.; Angrist, N.; Goldszmidt, R.; Kira, B.; Petherick, A.; Phillips, T.;
Webster, S.; Cameron-Blake, E.; Hallas, L.; Majumdar, S.; et al. A global panel
database of pandemic policies (Oxford COVID-19 Government Response
Tracker).
Nature Human Behaviour 2021, 5, 529-538,doi:10.1038/s41562-021-01079-8.
19. Hirvonen, K.; Bai, Y.; Headey, D.; Masters, W.A. Affordability of the EAT–
Lancet reference diet: a global analysis.
The Lancet Global Health 2020, 8, e59-e66, doi:https://doi.org/10.1016/S2214-109X(19)30447-4.
20. Khan, J.R.; Awan, N.; Islam, M.M.; Muurlink, O. Healthcare Capacity, Health Expenditure, and Civil Society as Predictors of COVID-19 Case Fatalities: A Global Analysis.
Frontiers in Public Health 2020, 8,
doi:10.3389/fpubh.2020.00347.
21. Rudd, K.E.; Johnson, S.C.; Agesa, K.M.; Shackelford, K.A.; Tsoi, D.; Kievlan, D.R.; Colombara, D.V.; Ikuta, K.S.; Kissoon, N.; Finfer, S.; et al. Global, regional, and national sepsis incidence and mortality, 1990–2017: analysis for the Global Burden of Disease Study.
The Lancet 2020, 395, 200-211, doi:https://doi.org/10.1016/S0140-6736(19)32989-7.

2. There is hardly a comparison with other studies that addressed the same
question. How do the estimates of the authors compare with the estimates of the
other publications?

Authors’ response:
Thanks for your kind reminders. At present, there has been a lack of studies linking national economic level, vaccination policy and vaccination
coverage. Some studies analyzed the impact of national economic level on
vaccination policy or vaccination coverage. We compared the results of our study with these studies. In the revised manuscript (tracks), we have added the statement to better
indicate this situation in the section of Discussion, line 232-244, page 7, as following:

Consistent with the results of previous studies, this study has shown that
national income level is an important factor influencing country-level vaccine access, vaccination policy, and vaccination coverage. Prior to widespread vaccine allocation by COVAX, a large number of studies expressed concern about the fact that the majority of global vaccine purchases and vaccinations were occurring in a few high income countries [7,26,27]. A study analyzing the relationship between macrosocioeconomic factors and the global allocation of COVID-19 vaccines showed that higher gross domestic product per capita was associated with larger numbers of vaccinations [28], and similar findings were also observed within the United States[29]. A few studies have focused on the different stages of vaccination policies indifferent countries, which showed that the poorest countries were often lagging behind in having clear strategies or resources to promote COVID-19 vaccination [30].However, there is still a lack of research to develop the relationship between national economic level, vaccination policy and vaccination coverage.”

3. The type of the study and the statistical approach should be mentioned in the
abstract. Authors’ response:
Thanks for your kind reminders. In the revised manuscript(tracks), we have added the information on the type of the study and the statistical approach in the section of Abstract, line 14-19, page 1, as following:

“Aiming to analyze the association between country income level and COVID

19 vaccination coverage and explore the mediating role of vaccination policy, we
conducted a cross-sectional ecological study. The dependent variable was COVID-19 vaccination coverage in 138 countries as of May 31, 2021. A single-mediator model based on structural equation modeling was developed to analyze mediation effects in different country income groups.”
Some more detailed comments:

4. Abstract: For all estimates in the abstract 95% CI should be provided (for the
regression coefficient as well as the effect of the vaccination policies.

Authors’ response:
Thanks for your kind reminders. In the revised manuscript
(tracks), we have added the 95% CI for the estimates in the section of Abstract, line19-24, page 1, as following:

“Compared with high-income countries, upper-middle- (β = -1.44, 95% CI: -
1.86- -1.02, p < .001), lower-middle- (β = -2.24, 95% CI: -2.67- -1.82, p < .001), and low- (β = -4.05, 95% CI: -4.59- -3.51, p < .001) income countries had lower vaccination coverage. Vaccination policies mediated 14.6% and 15.6% of the effect in upper-middle- (β = -0.21, 95% CI: -0.39- -0.03, p = .020) and lower-middle- (β = -0.35, 95% CI: -0.56- -0.13, p = .002) income countries, respectively, whereas the mediation effect was not significant in low-income countries (β = -0.21, 95% CI: -0.43- 0.01, p = .062).”

5. Material and Methods: Please provide more information about the quality of
the data.

Authors’ response:
Thanks for your kind reminders. In the revised manuscript (tracks), we have added the statement to reflect the quality of the data and referencesto evidence the data were widely used in scientific research in the section of Materials and Methods, line 106-107, page 3, as following: “We use these data for its reliability and consistent methods, which were widelyused in scientific research [17-21].”
17. Mathieu, E.; Ritchie, H.; Ortiz-Ospina, E.; Roser, M.; Hasell, J.; Appel, C.;
Giattino, C.; Rodés-Guirao, L. A global database of COVID-19 vaccinations.
Nature Human Behaviour 2021, 5, 947-953, doi:10.1038/s41562-021-01122-8.
18. Hale, T.; Angrist, N.; Goldszmidt, R.; Kira, B.; Petherick, A.; Phillips, T.;
Webster, S.; Cameron-Blake, E.; Hallas, L.; Majumdar, S.; et al. A global panel
database of pandemic policies (Oxford COVID-19 Government Response
Tracker).
Nature Human Behaviour 2021, 5, 529-538, doi:10.1038/s41562-021-01079-8.
19. Hirvonen, K.; Bai, Y.; Headey, D.; Masters, W.A. Affordability of the EAT–
Lancet reference diet: a global analysis.
The Lancet Global Health 2020, 8, e59-e66, doi:https://doi.org/10.1016/S2214-109X(19)30447-4.
20. Khan, J.R.; Awan, N.; Islam, M.M.; Muurlink, O. Healthcare Capacity, Health Expenditure, and Civil Society as Predictors of COVID-19 Case Fatalities: A Global Analysis.
Frontiers in Public Health 2020, 8,
doi:10.3389/fpubh.2020.00347.
21. Rudd, K.E.; Johnson, S.C.; Agesa, K.M.; Shackelford, K.A.; Tsoi, D.; Kievlan, D.R.; Colombara, D.V.; Ikuta, K.S.; Kissoon, N.; Finfer, S.; et al. Global, regional, and national sepsis incidence and mortality, 1990–2017: analysis for the Global Burden of Disease Study.
The Lancet 2020, 395, 200-211,doi:https://doi.org/10.1016/S0140-6736(19)32989-7.

6. Results: The differences observed in values of the regression coefficients
should be explained in relation to each other. The absolute values of the beta
estimates are not directly interpretable for the reader. The authors should
provide an interpretation of the beta values and a comparison.

Authors’ response: Thanks for your kind reminders. In the revised manuscript
(tracks), we have added the statement to provide an interpretation of the beta values in the section of Materials and Methods, line 156-161, page 4, as following:

“The correlation of path (a) represents the change in vaccination policy strength
in countries with other income levels compared with high-income countries; the
correlation coefficient of path (b) indicates the degree of change in national
vaccination coverage for each one-point of increase in policy strength. The product of (a) and (b), and the total effect coefficient, respectively, represent how country income level influences vaccination coverage by influencing vaccination policy strength and in general.”Moreover, we have expended the statement to explain the meaning of the results inthe section of Results, line 200-218, page 5-6, as following:

“In the regression analysis, the mediation models showed that income level was
significantly associated with vaccination coverage. Vaccination policy strength was significantly associated with income level and vaccination coverage, suggesting that vaccination policy strength was a mediator of country-level socioeconomic vaccination disparities. Specifically, compared with high-income countries, vaccination policy strength was lower in upper-middle- (β = -0.64, p = .007), lowermiddle- (β = -1.04, p <.001), and low- (β = -0.63, p = .042) income countries. Lower policy strength was associated with lower vaccination coverage (β = 0.33, p < .001). Compared with high-income countries, upper-middle- (β = -1.44, p < .001), lowermiddle- (β = -2.24, p <.001), and low- (β = -4.05, p < .001) income countries had
lower vaccination coverage. Vaccination policy strength explained 14.6% and 15.6% of the association between income level and vaccination coverage in upper-middle- (β = -0.21, p =.020) and lower-middle- (β = -0.35, p = .002) income countries, respectively. However, the mediation effect was not significant (β = -0.21, p = .062) in low-income countries. Similar results were observed after adjusting for demographic structure and underlying health conditions, demonstrating the robustness of the model(Table 2).”

7. Discussion: The quality of the data used and the implication of their crudeness
should be discussed.

Authors’ response: Thanks for your helpful suggestions. In the revised manuscript (tracks), we have expanded the statement to discuss the quality of the data and the implication of their crudeness in the section of Discussion, line 296-303, page 8, as following:

“This study has several limitations. First, because of limited data availability, the data used in this study could only represent country-level vaccination coverage to a certain extent, and could not reflect the specific situation at the sub-national level. The lack of data from low-income countries may have affected the accuracy of the results.
Second, because of the lack of specific data on vaccine deployment and vaccination acceptance, the model failed to incorporate multiple influencing factors, resulting in a simplification of the association. Finally, this was an ecological study and therefore could not assess causality.”

Reviewer 2 Report

Through regression analysis of publicly available data, the researchers explore the extent to which vaccination policy mediates the relationship between income and vaccine coverage across countries.  The overall finding is that countries with higher income levels tend to have more extensive polices and higher vaccination coverage.   

Suggestions for refinement:

Discussion Section: the authors assert that based on previous studies, vaccine acceptance tends to be lower in low income countries because of a lack of knowledge about disease risk, and the effectiveness of vaccines.  It would be good to consider expanding on their ideas of how to heighten knowledge about COVID-19 Risk as well as vaccine effectiveness.   

Conclusions section:  the authors might elaborate in 1-2 sentences on how to strengthen vaccine allocation frameworks. 

Author Response

Dear editor,
We appreciate the opportunity to respond the comments from the reviewers for our manuscript. We have revised the manuscript accordingly. The following are the point-to-point responses to these comments of Reviewer 1.
Comments and Suggestions for Authors:
The authors provide an analysis of effects of income and vaccine mediation policy on the vaccination coverage. The authors employ a statistical regression approach. The novelty of the manuscript is extremely limited.
1. The authors use data from different sources just providing the reference of the source. They may be more detailed about the type of the data and the quality of the data.
Authors’ response: Thanks for your valuable comments. In the revised manuscript (tracks), we have added a table (Table S2) to better show the type of data in the section of Appendix, line 337-342, page 9-10, as following:
 Table S2. Variables evaluated for testing in mediation model

Variable Definition Type Source Date of Data
Update
Dependent variable
COVID-19 vaccination
coverage
The total number of
vaccination doses
administered per 100
people at the country
level
Continuous
variable
Our World
in Data
From May 25
to 31, 2021
Independent variable

Country income level Four country
categories including
high-, upper-middle-,
lower-middle-, or
low-income
Categorical
variable
The World
Bank
2021
Mediation variable
Vaccination policy
strength
A score of 1–6
(lowest-highest
strength), which
incorporated the
population covered by
the country
vaccination policy, as
well as vaccine
affordability at the
country level
Continuous
variable
The Oxford
COVID-19
Government
Response
Tracker
From May 25
to 31, 2021
Covariates
Age 65 years The percentage of the
population ages 65
and above at the
country level
Continuous
variable
The World
Bank
2019
CVD prevalence The prevalence of
cardiovascular
diseases per 100
people at the country
level
Continuous
variable
The
Institute for
Health
Metrics and
Evaluation
2019
CRD prevalence The prevalence of
chronic respiratory
diseases per 100
people at the country
level
Continuous
variable
The
Institute for
Health
Metrics and
Evaluation
2019
DM prevalence The prevalence of
diabetes mellitus per
100 people at the
country level
Continuous
variable
The
Institute for
Health
Metrics and
Evaluation
2019

Note: The dependent variable and mediation variable used the latest data from May 25 to 31, 2021. Theindependent variable was mainly measured by the gross national income per capita in current UnitedStates dollars (using the World Bank’s Atlas method exchange rates) in 2019, with reference to the
World Bank’s income group categories for 2021. CVD: cardiovascular diseases; CRD: chronicrespiratory diseases; DM: diabetes mellitus.

Moreover, in the revised manuscript (tracks), we have added the statement to reflect the quality of the data and references to evidence the data were widely used in scientific research in the section of Materials and Methods, line 106-107, page 3, as following:

“We use these data for its reliability and consistent methods, which were widely used in scientific research [17-21].”
17. Mathieu, E.; Ritchie, H.; Ortiz-Ospina, E.; Roser, M.; Hasell, J.; Appel, C.; Giattino, C.; Rodés-Guirao, L. A global database of COVID-19 vaccinations. Nature Human Behaviour 2021, 5, 947-953, doi:10.1038/s41562-021-01122-8.
18. Hale, T.; Angrist, N.; Goldszmidt, R.; Kira, B.; Petherick, A.; Phillips, T.; Webster, S.; Cameron-Blake, E.; Hallas, L.; Majumdar, S.; et al. A global panel database of pandemic policies (Oxford COVID-19 Government Response Tracker). Nature Human Behaviour 2021, 5, 529-538, doi:10.1038/s41562-021-01079-8.
19. Hirvonen, K.; Bai, Y.; Headey, D.; Masters, W.A. Affordability of the EAT–Lancet reference diet: a global analysis. The Lancet Global Health 2020, 8, e59-e66, doi:https://doi.org/10.1016/S2214-109X(19)30447-4.
20. Khan, J.R.; Awan, N.; Islam, M.M.; Muurlink, O. Healthcare Capacity, Health Expenditure, and Civil Society as Predictors of COVID-19 Case Fatalities: A Global Analysis. Frontiers in Public Health 2020, 8, doi:10.3389/fpubh.2020.00347.
21. Rudd, K.E.; Johnson, S.C.; Agesa, K.M.; Shackelford, K.A.; Tsoi, D.; Kievlan, D.R.; Colombara, D.V.; Ikuta, K.S.; Kissoon, N.; Finfer, S.; et al. Global, regional, and national sepsis incidence and mortality, 1990–2017: analysis for the Global Burden of Disease Study. The Lancet 2020, 395, 200-211, doi:https://doi.org/10.1016/S0140-6736(19)32989-7.

2. There is hardly a comparison with other studies that addressed the same question. How do the estimates of the authors compare with the estimates of the other publications?

Authors’ response: Thanks for your kind reminders. At present, there has been a lack of studies linking national economic level, vaccination policy and vaccination coverage. Some studies analyzed the impact of national economic level on vaccination policy or vaccination coverage. We compared the results of our study with these studies. In the revised manuscript (tracks), we have added the statement to better indicate this situation in the section of Discussion, line 232-244, page 7, as following:

“Consistent with the results of previous studies, this study has shown that national income level is an important factor influencing country-level vaccine access, vaccination policy, and vaccination coverage. Prior to widespread vaccine allocation by COVAX, a large number of studies expressed concern about the fact that the majority of global vaccine purchases and vaccinations were occurring in a few high-income countries [7,26,27]. A study analyzing the relationship between macro-socioeconomic factors and the global allocation of COVID-19 vaccines showed that higher gross domestic product per capita was associated with larger numbers of vaccinations [28], and similar findings were also observed within the United States [29]. A few studies have focused on the different stages of vaccination policies in different countries, which showed that the poorest countries were often lagging behind in having clear strategies or resources to promote COVID-19 vaccination [30]. However, there is still a lack of research to develop the relationship between national economic level, vaccination policy and vaccination coverage.”

3. The type of the study and the statistical approach should be mentioned in the abstract.

Authors’ response: Thanks for your kind reminders. In the revised manuscript (tracks), we have added the information on the type of the study and the statistical approach in the section of Abstract, line 14-19, page 1, as following:

“Aiming to analyze the association between country income level and COVID-
19 vaccination coverage and explore the mediating role of vaccination policy, we conducted a cross-sectional ecological study. The dependent variable was COVID-19 vaccination coverage in 138 countries as of May 31, 2021. A single-mediator model based on structural equation modeling was developed to analyze mediation effects in different country income groups.”

Some more detailed comments:
4. Abstract: For all estimates in the abstract 95% CI should be provided (for the regression coefficient as well as the effect of the vaccination policies.

Authors’ response: Thanks for your kind reminders. In the revised manuscript (tracks), we have added the 95% CI for the estimates in the section of Abstract, line 19-24, page 1, as following:
“Compared with high-income countries, upper-middle- (β = -1.44, 95% CI: -1.86- -1.02, p < .001), lower-middle- (β = -2.24, 95% CI: -2.67- -1.82, p < .001), and low- (β = -4.05, 95% CI: -4.59- -3.51, p < .001) income countries had lower vaccination coverage. Vaccination policies mediated 14.6% and 15.6% of the effect in upper-middle- (β = -0.21, 95% CI: -0.39- -0.03, p = .020) and lower-middle- (β = -0.35, 95% CI: -0.56- -0.13, p = .002) income countries, respectively, whereas the mediation effect was not significant in low-income countries (β = -0.21, 95% CI: -0.43- 0.01, p = .062).”

5. Material and Methods: Please provide more information about the quality of
the data.

Authors’ response: Thanks for your kind reminders. In the revised manuscript (tracks), we have added the statement to reflect the quality of the data and references to evidence the data were widely used in scientific research in the section of Materials and Methods, line 106-107, page 3, as following:
“We use these data for its reliability and consistent methods, which were widely used in scientific research [17-21].”

17. Mathieu, E.; Ritchie, H.; Ortiz-Ospina, E.; Roser, M.; Hasell, J.; Appel, C.; Giattino, C.; Rodés-Guirao, L. A global database of COVID-19 vaccinations. Nature Human Behaviour 2021, 5, 947-953, doi:10.1038/s41562-021-01122-8.
18. Hale, T.; Angrist, N.; Goldszmidt, R.; Kira, B.; Petherick, A.; Phillips, T.; Webster, S.; Cameron-Blake, E.; Hallas, L.; Majumdar, S.; et al. A global panel database of pandemic policies (Oxford COVID-19 Government Response Tracker). Nature Human Behaviour 2021, 5, 529-538, doi:10.1038/s41562-021-01079-8.
19. Hirvonen, K.; Bai, Y.; Headey, D.; Masters, W.A. Affordability of the EAT–Lancet reference diet: a global analysis. The Lancet Global Health 2020, 8, e59-e66, doi:https://doi.org/10.1016/S2214-109X(19)30447-4.
20. Khan, J.R.; Awan, N.; Islam, M.M.; Muurlink, O. Healthcare Capacity, Health Expenditure, and Civil Society as Predictors of COVID-19 Case Fatalities: A Global Analysis. Frontiers in Public Health 2020, 8, doi:10.3389/fpubh.2020.00347.
21. Rudd, K.E.; Johnson, S.C.; Agesa, K.M.; Shackelford, K.A.; Tsoi, D.; Kievlan, D.R.; Colombara, D.V.; Ikuta, K.S.; Kissoon, N.; Finfer, S.; et al. Global, regional, and national sepsis incidence and mortality, 1990–2017: analysis for the Global Burden of Disease Study. The Lancet 2020, 395, 200-211, doi:https://doi.org/10.1016/S0140-6736(19)32989-7.

6. Results: The differences observed in values of the regression coefficients should be explained in relation to each other. The absolute values of the beta estimates are not directly interpretable for the reader. The authors should provide an interpretation of the beta values and a comparison.

Authors’ response: Thanks for your kind reminders. In the revised manuscript (tracks), we have added the statement to provide an interpretation of the beta values in the section of Materials and Methods, line 156-161, page 4, as following:

“The correlation of path (a) represents the change in vaccination policy strength in countries with other income levels compared with high-income countries; the correlation coefficient of path (b) indicates the degree of change in national vaccination coverage for each one-point of increase in policy strength. The product of (a) and (b), and the total effect coefficient, respectively, represent how country income level influences vaccination coverage by influencing vaccination policy strength and in general.”

Moreover, we have expended the statement to explain the meaning of the results in the section of Results, line 200-218, page 5-6, as following:

“In the regression analysis, the mediation models showed that income level was significantly associated with vaccination coverage. Vaccination policy strength was significantly associated with income level and vaccination coverage, suggesting that vaccination policy strength was a mediator of country-level socioeconomic vaccination disparities. Specifically, compared with high-income countries, vaccination policy strength was lower in upper-middle- (β = -0.64, p = .007), lower-middle- (β = -1.04, p <.001), and low- (β = -0.63, p = .042) income countries. Lower policy strength was associated with lower vaccination coverage (β = 0.33, p < .001). Compared with high-income countries, upper-middle- (β = -1.44, p < .001), lower-middle- (β = -2.24, p <.001), and low- (β = -4.05, p < .001) income countries had lower vaccination coverage. Vaccination policy strength explained 14.6% and 15.6% of the association between income level and vaccination coverage in upper-middle- (β = -0.21, p =.020) and lower-middle- (β = -0.35, p = .002) income countries, respectively. However, the mediation effect was not significant (β = -0.21, p = .062) in low-income countries. Similar results were observed after adjusting for demographic structure and underlying health conditions, demonstrating the robustness of the model (Table 2).”

7. Discussion: The quality of the data used and the implication of their crudeness should be discussed.

Authors’ response: Thanks for your helpful suggestions. In the revised manuscript (tracks), we have expanded the statement to discuss the quality of the data and the implication of their crudeness in the section of Discussion, line 296-303, page 8, as following:

“This study has several limitations. First, because of limited data availability, the data used in this study could only represent country-level vaccination coverage to a certain extent, and could not reflect the specific situation at the sub-national level. The lack of data from low-income countries may have affected the accuracy of the results. Second, because of the lack of specific data on vaccine deployment and vaccination acceptance, the model failed to incorporate multiple influencing factors, resulting in a simplification of the association. Finally, this was an ecological study and therefore could not assess causality.”

Reviewer 3 Report

Authors examine “ Disparities in COVID-19 Vaccination among Low-, Middle-, and High-Income Countries: The Mediating Role of Vaccination Policy” and report some useful findings such as vaccination coverage in 138 countries as of May 31, 2021. Using path model, a single-mediator model was developed to analyze mediation effects in different country income groups. The work is interesting but need to answer the following questions:

Further authors find that countries with higher income levels have taken the lead in implementing more comprehensive vaccination policies and have higher vaccination coverage. Its obvious high-income countries do high vaccination coverage and low will have lower. I do not find strong reason and rational for the is taking care during the model building and performing analysis.

What about Vaccination reluctance among 65+ age group0 in the emerging and low-income group, its not reflected in the (Y).

It’s unclear what the trimline start and end of the sampling, total cross-sectional observations, authors need to put one more table explaining the summary statistics of all variables.

There no discussion on the classification of 138 countries in sub groups according to income level, which approach or criterion used to group the countries.

There is no discussion of policy implications, how countries can enhance vaccine coverage looking at the demographics and social issues.

Author Response

Dear editor,
We appreciate the opportunity to respond the comments from the reviewers for our manuscript. We have revised the manuscript accordingly. The following are the point-to-point responses to these comments of Reviewer 3.
Reviewer Reports-Reviewer 3: Comments and Suggestions for Authors:
Authors examine “Disparities in COVID-19 Vaccination among Low-, Middle-, and High-Income Countries: The Mediating Role of Vaccination Policy” and report some useful findings such as vaccination coverage in 138 countries as of May 31, 2021. Using path model, a single-mediator model was developed to analyze mediation effects in different country income groups. The work is interesting but need to answer the following questions:
1. Further authors find that countries with higher income levels have taken the lead in implementing more comprehensive vaccination policies and have higher vaccination coverage. Its obvious high-income countries do high vaccination coverage and low will have lower. I do not find strong reason and rational for the is taking care during the model building and performing analysis.

Authors’ response: Thanks for your valuable comments. In the revised manuscript (tracks), we have revised the statement to explain the significance of this study in the section of Introduction, line 77-89, page 2, as following:
“In summary, under current global allocation framework of vaccines, country’s income levels will affect its vaccine access, which in turn affects its vaccine deployment and ultimately vaccination coverage. Previous studies on COVID-19 vaccine equity have mostly focused on vaccine access in countries with different income levels, without considering country-level vaccine deployment (i.e., domestic vaccination policies). This study analyzed the impact of national income level on vaccination policies and vaccination coverage and explored the mediating role of vaccination policies in the relationship between a country’s income level and vaccination coverage. Therefore, we can break down the different stages of national vaccine access and deployment to deeply explore the factors that influence vaccination coverage.”

2. What about Vaccination reluctance among 65+ age group0 in the emerging and low-income group, its not reflected in the (Y).
Authors’ response: Thanks for your kind reminders. We agree that vaccination among 65+ age group in the emerging and low-income group is a very important factor influencing the vaccine coverage. However, due to data limitation, we had no data on individual country’s vaccine acceptance by age, to investigate the impact of this factor on the vaccine coverage. The impact was included in the direct effects of the model and we discussed it in the limitations of the study. In the revised manuscript (tracks), we have added statements to further emphasize this part in the section of Discussion, line 273-280, page 7, as following:
“However, several studies have also shown that vaccine acceptance is lower among low-income populations because of a lack of accurate knowledge about COVID-19 risks and vaccine effectiveness [35-39]. In particular, vaccination reluctance among 65+ age group in the emerging and low-income group is more noteworthy. Vaccine deployment at the country level and acceptance at the individual level may be influencing factors that were not included in the model in this study, and differences in these factors may be part of the reason why vaccine policies did not show a significant mediation effect in low-income countries.”
And in the section of Discussion, line 299-302, page 8, as following:
“Second, because of the lack of specific data on vaccine deployment and vaccination acceptance, the model failed to incorporate multiple influencing factors, resulting in a simplification of the association.”

3. It’s unclear what the trimline start and end of the sampling, total cross-sectional observations, authors need to put one more table explaining the summary statistics of all variables.
Authors’ response: Thanks for your helpful suggestions. In the revised manuscript (tracks), we have added a table (Table S2) to summarize statistics of all variables in the section of Appendix, line 337-342, page 9-10, as following:
 Table S2. Variables evaluated for testing in mediation model

Variable Definition Type Source Date of Data
Update
Dependent variable

COVID-19 vaccination
coverage
The total number of
vaccination doses
administered per 100
people at the country
level
Continuous
variable
Our World
in Data
From May 25
to 31, 2021
Independent variable
Country income level Four country
categories including
high-, upper-middle-,
lower-middle-, or
low-income
Categorical
variable
The World
Bank
2021
Mediation variable
Vaccination policy
strength
A score of 1–6
(lowest-highest
strength), which
incorporated the
population covered by
the country
vaccination policy, as
well as vaccine
affordability at the
country level.
Continuous
variable
The Oxford
COVID-19
Government
Response
Tracker
From May 25
to 31, 2021
Covariates
Age 65 years The percentage of the
population ages 65
and above at the
country level
Continuous
variable
The World
Bank
2019
CVD prevalence The prevalence of
cardiovascular
diseases per 100
people at the country
level
Continuous
variable
The
Institute for
Health
Metrics and
Evaluation
2019
CRD prevalence The prevalence of
chronic respiratory
diseases per 100
people at the country
level
Continuous
variable
The
Institute for
Health
Metrics and
Evaluation
2019
DM prevalence The prevalence of
diabetes mellitus per
100 people at the
country level
Continuous
variable
The
Institute for
Health
Metrics and
Evaluation
2019

Note: The dependent variable and mediation variable used the latest data from May 25 to 31, 2021. The independent variable was mainly measured by the gross national income per capita in current United States dollars (using the World Bank’s Atlas method exchange rates) in 2019, with reference to the World Bank’s income group categories for 2021. CVD: cardiovascular diseases; CRD: chronic respiratory diseases; DM: diabetes mellitus.

4. There no discussion on the classification of 138 countries in sub groups according to income level, which approach or criterion used to group the countries.
Authors’ response: Thanks for your kind reminders. We indicated the criterion used to group the countries in the section of Measures (line 118-124, page 3), and notes of tables as following:
“The independent variable was the income level of each country. This was mainly measured by the gross national income per capita in current United States dollars (using the World Bank’s Atlas method exchange rates) in 2019 for each country, and all included countries were categorized as high-, upper-middle-, lower-middle-, or low-income, with reference to the World Bank’s income group categories for 2021 [22]. Among 138 countries included for analysis, there were 51 high-income countries, 36 upper-middle-income countries, 34 lower-middle-income countries, and 17 low-income countries”

5. There is no discussion of policy implications, how countries can enhance vaccine coverage looking at the demographics and social issues.
Authors’ response: Thanks for your helpful suggestions. To reflect the policy implications, in the revised manuscript (tracks), we have added relevant information in the section of Discussion, line 281-295, page 8, as following:
“Based on previous studies [35-39], vaccine acceptance tends to be lower in low-income countries because of a lack of knowledge about disease risk and the effective-ness of vaccines. It is warranted to expand on their ideas of how to heighten knowledge about COVID-19 risk as well as vaccine effectiveness. Countries need to make efforts to shape scientific awareness and attitudes towards COVID-19 vaccines. Behavioral demonstration by cultural or public health leaders, health education by primary care personnel would make a difference. Furthermore, long-term efforts are still needed. Especially in the middle and late stages of global allocation of COVID-19 vaccines, the decreasing urgency of some high-income countries for their own vaccination needs may increase their willingness to promote vaccine equity. Strong advocacy by key actors in multilateral, cross-regional and regional mechanisms, as well as relatively binding benefit sharing and regulatory mechanisms will play a catalytic role. Strengthening vaccine deployment capacity in low- and middle-income countries should also be emphasized. On the basis of funding, equipment and technical support, it would be helpful to mobilize practice teams to help these countries in the last-mile deployment.”

Round 2

Reviewer 3 Report

Authors have improved the work, now it can be considered for further editorial considerations